# Social distancing with chronic pain during COVID-19: A cross-sectional correlational analysis

**Bethany Donaghy** [ID]*[Ͻ], **Susannah C. Walker**[Ͻ], **David J. Moore** [ID][Ͻ]

School of Psychology, Liverpool John Moores University, Liverpool, England

Ͻ These authors contributed equally to this work.
* B.E.Donaghy@2016.ljmu.ac.uk

## Abstract

### Background

Understanding of the role social factors play in chronic pain is growing, with more adaptive and satisfying social relationships helping pain management. During the COVID-19 pandemic, social distancing measures facilitated a naturalistic study of how changes to social interaction affected chronic pain intensity.

### Methods

In a cross-sectional correlational design, questionnaire data was collected over a 38-day period during the March 2020 COVID-19 lockdown, individuals with chronic pain were asked about their current pain experience as well as notable social factors which might relate to pain.

### Results

Multiple regression analysis revealed social satisfaction significantly predicted pain experience, with a reduction in social participation during COVID-19 lockdowns increasing pain disability, and increased social satisfaction associated with decreasing pain intensity.

### Conclusions

While pain management often focuses on the functional aspects of pain alleviation, these findings suggest psychological aspects of socialising satisfaction also impact pain experience. Pain management strategies should consider ways to increase social satisfaction in individuals with chronic pain, perhaps by facilitating socialisation in the home using remote communication methods similar to those which became popular during the COVID-19 lockdown.

**Data Availability Statement:** A link to our data repository can be find via this link: https://doi.org/10.24377/LJMU.d.00000132.

**Funding:** The author(s) received no specific funding for this work.

**Competing interests:** The authors have declared that no competing interests exist.

## Introduction

The COVID-19 pandemic of 2020 changed the social landscape of how people live, work, and interact with sources for support [1, 2]. Guidelines presented by most governments and international health experts included reduced public transport use, closure or reduced operation of restaurants and bars, and the requirement to distance ourselves from those we are not living with [3]. The mental health impact of such distancing measures have been discussed at length, emphasising the need for high quality eHealth [4, 5]. Furthermore, whilst pre-lockdown baseline data suggest individuals with chronic pain may perceive their pain to be significantly worsened under the lockdown, pain reports remain unchanged [6]. This suggests individuals with chronic pain have greater lockdown anxiety than non-pain controls, and pain catastrophizing may predict lockdown pain experience. However, these studies have not considered the role of changing social interactions on chronic pain.

Lack of social interaction due to functional limitations can cause those with chronic pain to feel socially isolated, in turn increasing their pain experiences [7]. Moreover, difficulties with social relatedness impact pain, whereby other people's behaviours can affect pain experience [8]. For example, parental displays of pain-promoting behaviour such as pain reassurance have been shown to increase child pain intensity [9], whilst supportive behaviour such as flexibility to pain may decrease child pain intensity [10]. The latter is demonstrated further by the fact 's the behaviour of friends/family can have both positive and negative effects on those suffering chronic pain [11, 12]. For example, whilst pain catastrophizing can increase pain experience, positive instrumental support, like preparing meals, can decrease individuals' perception of pain. Thus, further emphasising how behaviours of those the chronic pain population socialises with affects their pain experience.

For many individuals with chronic pain some degree of limited social interaction was the norm prior to COVID-19 [13], therefore feelings of social isolation may decrease with increased interaction with those we cohabit with [14], and reduced interactions outside of the home; a concept often theorised, but not yet tested [15]. Studies have shown perceived suffering of those with chronic pain may have increased during lockdown [6] however roles of social buffers in mitigating this distress have not yet been explored.

The present study aimed to understand how social distancing measures implemented during the 2020 COVID-19 pandemic socially impacted those with chronic pain, helping further understanding of social factors in pain. The research considers how we may buffer pain in future periods of isolation; whereby social risk factors, such as loneliness, may increase chronic pain intensity [15]. It is hypothesised all social factors (loneliness, satisfaction with participation in social roles, and difference in ability to participate in social roles and activities pre- and post-COVID-19) will significantly predict variance in pain and pain disability.

## Methods

### Participants

Over a 38-day span throughout March and April 2020 international Covid-19 lockdowns, a snowball sample of 122 participants were recruited through social media advertisements shared on both researcher and pain organisations' Facebook, Twitter, and LinkedIn accounts. All participants satisfied inclusion criteria of being aged 18+; having at least one chronic pain diagnosis or having experienced consistent pain for the past 6 months at time of recruitment, thus satisfying chronic pain diagnostic criteria [16]; having English as a first language, or ability to comprehend English at a level to understand questions; and no severe condition affecting ability to read or answer questions.

## Materials

Eight questionnaires were used to collect data within 3 domains: demographics and pain experience, home environment, and social environment. Data provided insight into the experiences of individuals with chronic pain, whilst social distancing during the COVID-19 pandemic.

**Demographics and pain experience.** *Demographics and Pain*. Participants provided standard demographic information, including gender and age. Further information was collected regarding participant pain diagnoses, pain duration, possible fears surrounding COVID-19, and broad social distancing information, such as length of time since participant began social distancing.

*PROMIS pain interference*. A 23-item version of the 40-item bank was used (adjusted for question relevancy) measuring day-to-day pain interference over the previous 7 days. Using a 5-point Likert scale ranging from 0 (not at all) to 5 (very much), participants rated 23 statements asking how pain interfered with their ability to, for example, take in new information or enjoy life. Higher overall score suggested pain interfered highly with participants' day-to-day life.

*Pain Disability Index (PDI)*. The PDI asks participants to rate on a 11-point intensity scale ranging from 0 (no disability) to 10 (worst disability), how much their pain disrupted their lives in 7 domains: family/home responsibilities; recreation; social activity; occupation; sexual behaviour; self-care; and life-supporting activities. Higher rating indicated a greater pain disability in the subsequent domain.

*Pain intensity*. A 3-item bank measuring participants' retrospective, and current pain intensity, providing data on average pain intensity 7-days prior, worst pain intensity 7-days prior, and pain at time of participation. Participants rated 3 statements regarding how intense they perceived their pain, using a free scale slider, anchored from 0 (no pain) on the left most side to 100 (worst pain imaginable) on the right most side. Higher scores indicated higher pain intensity.

Statements 1 and 2 were retrospective (participants indicated pain intensity in the past 7 days), and statement 3 was current (participants indicated pain intensity at time of participation).

**Home environment.** *PROMIS instrumental support–Calibrated items v2.0*. 11-item bank measuring instrumental support designed to gain understanding of how participants are functionally supported by others. Participants rated how often they had available, or received, help from others using a 5-point Likert scale ranging from 1 (never) to 5 (always) at 1-point intervals. Higher scores indicated more frequent instrumental support.

**Social environment.** *NIH toolbox loneliness scale v2.0 18+*. A 5-item bank measuring loneliness during social distancing. Using a 5-point Likert scale ranging from 1 (never) to 5 (always), participants rated how often in the past month they related to 5 statements. A higher score indicated the participant felt increasingly lonely.

*PROMIS satisfaction with participation in social roles and activities–Calibrated items v1.0*. A 14-item bank measuring participants' satisfaction with their participation in social roles because of their chronic pain diagnoses. Reflecting on the past 7 days, participants rated using a 5-point Likert scale ranging from 1 (not at all) to 5 (very much), 14 statements asking about their social participation satisfaction. A higher total score indicated a higher satisfaction with participation in social roles.

*PROMIS ability to participate in social roles and activities v2.0*. A 35-item bank measuring how pain limits participants' ability to participate in social roles and activities. Using a reversed 5-point Likert scale ranging from 5 (never) to 1 (always), participants rated 35 statements

regarding their ability to participate socially. The question bank was answered twice in two different contexts: first in the context of pre-COVID-19 lockdown initiation, second post-COVID-19 lockdown initiation. A higher score indicated a lower ability to participate in social roles and activities.

## Procedure

Following ethical approval from Liverpool John Moores University (LJMU) Psychology Research Ethics Committee (PSYREP), participants were contacted through social media adverts and provided with a study link which redirected to the study's host site, Qualtrics [17] for participation. Participants read a participation information sheet (PIS) which explained the study, informed them that study completion would be taken as implied consent, confirmed data was anonymised, and thus withdrawal rights were only applicable up until study completion.

Thereafter, participants completed a set of online questionnaires split into three subsections:

1. Demographics and experience of pain

2. Home environment

3. Social environment

Upon questionnaire completion, participants were debriefed with additional contact details for outlets providing support with pain, and COVID-19. Following debrief, participants were able to close the browser with the understanding data collected was now final and unable to be withdrawn.

## Data cleaning

Three participants were removed as when asked "if there was anything to add", all indicated they became bored during the study and felt some questions were irrelevant. Thus, suggesting answers may have been forced or not representative of their COVID-19 experiences and therefore unable to provide a valid understanding of how the participant had been impacted during COVID-19 social distancing measures. One further participant was removed as their diagnosis did not satisfy chronic pain diagnostic criteria. Therefore, a final dataset of 86 participants was used for analysis.

## Statistical analysis

Pre-processed data was analysed using IBM SPSS Statistics 25 [18]. Descriptive statistics for each variable were calculated, and a statistical significance of $p < .05$ was set for all inferential tests [19].

Five multiple linear regressions were performed respectively for each of the 5 outcome variables, each using all 4 predictor variables. Data was checked against multiple linear regression assumptions, with assumptions being tenable for analysis. Adjusted $R^2$ indicated variability ranged from 11.50–45.50% amongst predictors, and no multicollinearity issues were identified as correlations were linear, and all VIF statistics fell under $<5$. Furthermore, Cook's distance was substantially less than one for all variables indicating no unduly influential cases. Standardized residuals fell between -3 and +3 for pain interference, average 7-day pain intensity, and pain intensity at time of participation, indicating no issues of outliers. Standardized residuals for PDI, and worst 7-day pain intensity marginally exceeded the -3-minimum, suggesting

**Table 1. Participants chronic pain diagnostic information.**

| Diagnosis | Number of Participants Diagnosed |
|---|:---:|
| Complex Regional Pain Syndrome (CRPS) | 30 |
| Lower Back Pain | 24 |
| Fibromyalgia | 21 |
| Other[a] | 118 |

Note. Total number of participants diagnosed exceeds, *n* = 90, as participants can select multiple diagnoses.
[a]See S1 Table. for full list of diagnoses categorised as "other".

possible outliers. However, as all residuals were normally distributed and random, there were no issues of outliers and multiple linear regressions were deemed acceptable for analysis.

## Results

Of the 122 participants, 90 completed all elements of the survey and were thus eligible for inclusion prior to data cleaning. Participants consisted of 5 males, 83 females, and 2 non-binary individuals, aged between 21 and 65 years (*M* = 39.08, *SD* = 12.14). Thus, this sample was reflective of the chronic pain population [20]. At time of survey completion, participants were residing in the UK (72.40%), USA (15.50%), Australia (4.40%), Canada (3.30%), Spain (2.20%), Netherlands (1.10%), and Sweden (1.10%). All of these countries experienced similar social distancing measures at study commencement, however measures in Sweden appeared less stringent with social distancing deemed as a recommendation rather than a compulsion [21]. At time of participation, 43.30% of participants had at most one chronic pain diagnosis, 20% had at most two chronic pain diagnoses, and 36.70% had at least 3 chronic pain diagnoses. See Table 1. for further participant chronic pain diagnoses information.

Mean values and standard deviations of the final sample (*n* = 86) for all variables can be found in Table 2. These data show that our population appears to reflect similar pain and psychological profiles of other COVID-studies conducted with chronic pain populations [6]. Despite observing a marginal decrease in pain in intensity at time of participation (*M* = 56.02, *SD* = 23.07), relative to average pain intensity 7-days prior to participation (*M* = 57.68, *SD* = 19.73), this change was not statistically significant (*p* = .33).

To examine how social factors predicted pain during the first COVID-19 lockdown, social variables (instrumental support, loneliness, difference is social roles pre-lockdown-date, social

**Table 2. Mean values and standard deviations for all outcome and predictor variables.**

| | Mean (Standard Deviation) |
|:---:|:---:|
| PDI | 45.28 (14.06) |
| Pain Interference | 87.99 (17.73) |
| Average Pain Intensity 7-days Prior | 57.68 (19.73) |
| Worst Pain Intensity 7-days Prior | 78.84 (17.91) |
| Pain Intensity at Time of Participation | 56.02 (23.07) |
| Instrumental Support | 41.48 (10.98) |
| Loneliness | 15.34 (4.95) |
| Difference in Ability to Participate in Social Roles and Activities Pre- and Post-COVID-19 lockdown initiation | -14.66 (25.30) |
| Satisfaction with Participation in Social Roles and Activities | 28.01 (11.67) |

**Table 3. Multiple linear regression values for all outcome and predictor variables.**

| Outcome Variable | $R^2$ Value | Predictor Variable | $B$ | Standardized $\beta$ | $t$-value |
|---|---|---|---|---|---|
| Pain Interference | .16 | Instrumental Support | -.11 | -.07 | .70 |
| | | Loneliness | 1.07 | .30 | 3.10* |
| | | Difference in Ability to Socially Participate Pre-/Post-COVID-19 | .11 | .15 | 1.84 |
| | | Satisfaction with Social Participation | -.79 | -.52 | 6.16** |
| PDI | .46 | Instrumental Support | -.10 | -.08 | .79 |
| | | Loneliness | .01 | .002 | .02 |
| | | Difference in Ability to Socially Participate Pre-/Post-COVID-19 | .12 | .22 | 2.54* |
| | | Satisfaction with Social Participation | -.79 | -.66 | 7.59** |
| Average pain intensity 7-days prior | .25 | Instrumental Support | -.25 | -.14 | 1.20 |
| | | Loneliness | .05 | .01 | .10 |
| | | Difference in Ability to Socially Participate Pre-/Post-COVID-19 | .10 | .13 | 1.30 |
| | | Satisfaction with Social Participation | -.77 | -.46 | 4.44** |
| Worst pain intensity 7-days prior | .16 | Instrumental Support | -.02 | -0.1 | .08 |
| | | Loneliness | .22 | .06 | .49 |
| | | Difference in Ability to Socially Participate Pre-/Post-COVID-19 | .08 | .11 | 1.04 |
| | | Satisfaction with Social Participation | -.57 | -.38 | 3.41** |
| Pain intensity at time of participation | .30 | Instrumental Support | -.23 | -.11 | .94 |
| | | Loneliness | -.04 | -.01 | .08 |
| | | Difference in Ability to Socially Participate Pre-/Post-COVID-19 | -.02 | -.03 | .27 |
| | | Satisfaction with Social Participation | -1.04 | -.52 | 5.18** |

Note.

*Significant at $p < .05$

**Significant at $p \le .001$

role satisfaction and instrumental support) were entered to predict pain outcomes (pain interference, pain disability, 7-day average pain intensity, 7-day worst pain intensity, and pain intensity at time of participation) using enter method multiple regression analyses (see Table 3). Scores on the pain interference measure were significantly predicted by social factors, $F(4, 81) = 18.73$, $p < .001$ $R^2 = .16$. Examination of the individual predictors revealed greater reports of loneliness significantly predicted greater pain interference ($t = 3.10$, $p = .003$) while increases in social role satisfaction significantly predicted lower pain interference ($t = 6.16$, $p < .001$). Scores on the PDI were also significantly predicted by social factors $F(4, 81) = 16.93$, $p < .001$, $R^2 = .16$. Examination of the individual predictors shows that a greater reduction in social participation during COVID-19 lockdowns predicted increased pain disability ($t = 2.54$, $p = .01$) while increases in social role satisfaction predicted lower pain disability ($t = 7.59$, $p < .001$).

Further, all three pain intensity variables were significantly predicted by social factors: 7-day average pain $F(4, 77) = 6.45$, $p < .001$, $R^2 = .25$, 7-day worst pain $F(4, 77) = 3.46$, $p = .009$, $R^2 = .16$ and pain intensity at time of participation $F(4, 77) = 8.33$, $p < .001$, $R^2 = .30$. Examination of individual predictors shows that only social role satisfaction predicted pain intensity; average pain ($t = 4.44$, $p < .001$), worst pain ($t = 3.41$, $p = .001$), current pain ($t = 5.18$, $p < .001$).

These data support the hypothesis that satisfaction with social roles is the strongest predictor of both pain intensity and the impact of pain. The impact of pain is also related to reductions in social role availability and loneliness, suggesting that other more structural variables are important here.

## Discussion

The primary purpose of this article was to explore the role of social factors related to pain in predicting the response of individuals with chronic pain to the COVID 19 pandemic and to the distancing measures put in place by governments to combat the spread of this virus. Previous research has established that those with chronic pain may be suffering more pain as a consequence of lockdown [6]. Our data indicate that many living with chronic pain were more satisfied with their ability to socialise during, compared to prior to, the lockdown. We found a significant reduction in perceived ability to participate in social activities compared to pre the lockdown period, suggesting that individuals with chronic pain, like most of the population, had their social worlds impacted by lockdown. Importantly these data suggest that individuals who had the greatest satisfaction with their social roles during lockdown reported the lowest levels of pain intensity, and pain related negative impacts on their lives. These data also suggest that self-reported loneliness and increased social relatedness since lockdown might be important in understanding functioning during the COVID pandemic.

Outside the context of COVID 19 lockdowns, an extensive narrative review highlighted chronic pain causes negative effects upon socialising abilities, with those with chronic pain socialising less and feeling more socially isolated than others in society [7]. Yet, despite suggested negative social experiences accompanying chronic pain, our findings further emphasise that social factors may positively alter pain experience [7, 14, 22–24]. For example, whilst at home with the family or friends available during COVID, individuals may have benefited from the social presence of others, thus supporting Edwards, Eccleston [22] and reinforcing the mere presence of a friend, or family member can alleviate chronic pain. These findings also highlight the importance of understanding broader family dynamics and how, for example, support from family members might alter pain perception. Therefore, we further recommend a holistic approach when considering long term pain management, considering the role of family members in altering an individual's pain.

Our most consistent finding here was that satisfaction with participation in social roles and activities may reduce perceptions of chronic pain. These findings replicate that of Solé, Racine [25] whom considered the roles social factors play in pain interference and depressive symptoms, and found increased satisfaction with social roles and activities, and self-perceived ability to participate socially significantly predicted a decrease in pain (particularly pain interference). Thus, our findings further highlight the importance of satisfactory socialising abilities within pain, as opposed to the functional ability to socialise itself. However, such observations arguably contradict the findings of Edwards, Eccleston [22] which suggests presence of an individual with whom a relationship is established, whether that be friend or partner, may be a key driver to reducing pain experience. The latter findings lend support to the notion that the functional abilities of socialising may reduce pain. This may require a level of active involvement on the individual with chronic pain's behalf, such as travel. However, our findings suggest that merely adhering to the emotional aspect of socialising, specifically through satisfaction, can reduce pain without the need for active involvement such as going to a restaurant. Whilst it is clear from both studies that socialising is an important aspect to reducing experiences of pain, by adjusting support to implement the emotional aspect of social satisfaction, barriers usually posed through functional aspects of chronic pain can be avoided, providing an alternative way to socialise and as a result, reduce pain experience. This finding is particularly important in the current technological revolution whereby individuals are more connected than ever through, for example, social media outlets providing further options as to how satisfaction with social roles can be incorporated into homes to effectively reduce chronic pain [26]. Therefore, it is plausible if an individual is unable to leave their house at any given

point due to their pain intensity, rather than increasing functional abilities to actively socialise by physically going to a restaurant with someone helping the individual travel safely, for example, ways to bring social satisfaction into the immediate home, through technology may be an effective alternative to change chronic pain experiences. However, given the heterogeneity of the current sample, efficacy of the latter may differ dependent on, for example, pain condition, age, gender, etc.

Currently, instrumental support remains one of the most widely used methods of social pain management in occupational therapy, whether that be through help from others, or home adjustments such as raised seats. Indeed, instrumental support interventions are effective at alleviating chronic pain and improving pain functionality [27–29]. However, the current data highlights those emotional aspects of social support are also important in the pain experience of individuals with chronic pain. Thus future interventions should consider social satisfaction alongside instrumental support, as this is consistent with previous findings that individuals with chronic pain who were satisfied with both their social, and instrumental support, showed greater improvements in pain experiences than those satisfied with just one aspect (for example QoL) [30, 31]. Thus, future interventions should integrate social satisfaction considerations within pain management approaches as this seems more likely to alleviate pain than instrumental support alone.

Although our findings suggest socialising can reduce the negative impact of chronic pain experience during COVID-19, evidence is lacking to validate the efficacy of social interactions in improving chronic pain experience. As the first pandemic of the current generation, evidence was not available to select variables vital in gaining an in-depth understanding of chronic pain during COVID-19. Instead, variables used in previous social interaction and pain research conducted in 'typical' social climate were used. For example, it is now understood COVID-19 caused a significant increase in stress [32]–a factor found to significantly alter how an individual socialises [33], and perceives pain [34]. Therefore, whilst satisfaction with ability to socialise was important to improving chronic pain experience, further variables, such as stress, may have provided further insight into factors predicting pain experience during lockdown. Moreover, variables such as loneliness and social satisfaction were only measured at time of study participation. Therefore, improvements or detriments to loneliness and social satisfaction as a consequence of the pandemic lockdown cannot be determined.

Furthermore, use of a global population may have posed limitations as although generally countries around the world were following similar lockdown timescales, some countries were lifted restrictions faster than others. For example, the UK remained in lockdown during May 2020 whilst Spain had begun to ease social distancing measures. Moreover, whilst chronic pain was defined in inclusion criteria as being either a diagnosis, or recurring pain for at least 6 months; by conducting the study online no formal assessment was used to satisfy inclusion criteria, for example, medical records to determine if pain was chronic/diagnosed. Therefore, it is likely a small percentage of participants are not representative of the chronic pain population as they perceived they have chronic pain when they do not (possibly due to low pain threshold's) or have pain which feels recurring for 6 months due to pain's ability to dilate time perception [35]. Thus, this sample may represent skewed socialising abilities due to lockdown socialising discrepancies and may not have represented the chronic pain population. Additionally, use of snowball sampling may have contributed to potential recruitment bias given a network of individuals who lead back to the principal investigator are targeted, creating a sample which by definition cannot be entirely random and fully representative of the chronic pain population [36].

Going forward, it will be beneficial to consider the social impact of COVID-19 on chronic pain longitudinally, examining how the changing nature of and attitudes towards socialising,

both during times of social distancing and when more typical social interactions resume, impact chronic pain responses. We propose that studies should focus on how social satisfaction can be improved using technology and how satisfaction with social roles might be utilised through social prescribing to support those living in pain. Through such focus, validity of social interactions in improving the experience of pain in both a possible second wave, and in everyday life can be improved and thus considered regularly in pain management.

By studying how individuals with chronic pain were impacted by COVID-19 lockdown, it is concluded that considerations of satisfaction with socialising abilities are effective at altering experiences of chronic pain. It should be noted further that as such satisfaction is subjective to an individual, integrating socialising satisfaction into current pain management presents an individualistic, rather than generalised, chronic pain management approach. However, further research is required to understand how such effects upon chronic pain experience may change when social climates return to normal to further validate the importance of social satisfaction when managing pain.

## Supporting information

**S1 Table. List of all diagnoses–including a breakdown of others.**
(TIF)

## Acknowledgments

Liverpool John Moores University (LJMU) Psychology Research Ethics Panel (PSYREP) provided ethical approval for this study

## Author Contributions

**Conceptualization:** Bethany Donaghy, Susannah C. Walker, David J. Moore.

**Data curation:** Bethany Donaghy, Susannah C. Walker, David J. Moore.

**Formal analysis:** Bethany Donaghy, David J. Moore.

**Investigation:** Bethany Donaghy, Susannah C. Walker, David J. Moore.

**Methodology:** Bethany Donaghy, Susannah C. Walker, David J. Moore.

**Project administration:** Bethany Donaghy, Susannah C. Walker, David J. Moore.

**Resources:** Bethany Donaghy, David J. Moore.

**Supervision:** Susannah C. Walker, David J. Moore.

**Writing – original draft:** Bethany Donaghy, David J. Moore.

**Writing – review & editing:** Bethany Donaghy, Susannah C. Walker, David J. Moore.

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
