## [Decision Letter · Decision Letter 0]

19 Jul 2022

PONE-D-22-02463Social distancing with chronic pain during COVID-19: a cross-sectional correlational analysis.PLOS ONE

Dear Dr. Donaghy,

Thank you for submitting your manuscript to PLOS ONE. After careful consideration, we feel that it has merit but does not fully meet PLOS ONE’s publication criteria as it currently stands. Therefore, we invite you to submit a revised version of the manuscript that addresses the points raised during the review process.

The manuscript has been evaluated by three reviewers, and their comments are available below.

The reviewers have raised a number of concerns that need attention. They request additional information on methodological aspects of the study and they question the conclusions drawn from the results.

Could you please revise the manuscript to carefully address the concerns raised?

We look forward to receiving your revised manuscript.

Kind regards,

Thomas Tischer

Staff Editor

PLOS ONE

Journal Requirements:

Additional Editor Comments:

Please provide the questionnaire(s) and coding key (if applicable)Please provide information how the questionnaire(s) were developed and validated

Reviewers' comments:

Reviewer's Responses to Questions

**Comments to the Author**

1. Is the manuscript technically sound, and do the data support the conclusions?

Reviewer #1: Yes

Reviewer #2: Yes

Reviewer #3: Partly

2. Has the statistical analysis been performed appropriately and rigorously? 

Reviewer #1: Yes

Reviewer #2: Yes

Reviewer #3: Yes

3. Have the authors made all data underlying the findings in their manuscript fully available?

Reviewer #1: Yes

Reviewer #2: Yes

Reviewer #3: Yes

4. Is the manuscript presented in an intelligible fashion and written in standard English?

Reviewer #1: No

Reviewer #2: Yes

Reviewer #3: No

5. Review Comments to the Author

Reviewer #1: [GRAMMATICAL ERRORS]

Line 99:

"Thus, was reflective of the chronic pain population".  the subject is missing in this sentence. Maybe correct like this: "Thus, this sample was...".

Line 230:

"To examine for how social factors predicted pain during the first COVID-19 lockdown social variables...".  change sentence: "To examine how social factors predicted pain during the first COVID-19 lockdown, social variables...".

Line 263:

"Importantly these confirm that those individuals who...".  the subject is missing in this sentence. Did the author mean: "Importantly this data confirms that the individuals who...".

Line 312: there is a question mark that shouldn't be there.

[OTHER REVIEWS]

#1 (regarding the abstract)

I would suggest taking the last sentence of the results (line 41-43) and including it in the conclusions. After that I would suggest developping a bit more the results.

#2 (regarding the methods)

Line 93-94: "...having experienced consistent pain for the past 6 months..."  maybe specify if was there a cut-off for pain intensity in this criteria.

#3 (regarding the results)

Line 222-224: "In our study we observe a marginal decrease in average pain intensity 7-days prior to (...) and pain intensity at the time of participation..."  marginal decrease compared to what? Is it between average pain intensity and pain intensity at the time of participation?

In that case I suggest modifying it as follows: "In our study we observe a marginal decrease between average pain intensity 7-days prior to (...) and pain intensity at the time of participation...".

Also, although it's descriptive statistics, it should be specified whether this difference is statistically significant or not.

#4 (regarding the conclusions)

Beware that the chronic pain population of this study is fairly heterogeneous (as shown by the diagnostic table S1). This should be noted in the conclusions and taken into consideration when stating recommendations/suggestions or making conclusions (such as Line 277-278 and Line 292-295).

Reviewer #2: The authors present an extremely well written manuscript an the effect of social distancing on people with chronic pain during the COVID lockdown. The COVID pandemic and the social restrictions provided an situation which was unprecedented before. Of course, this provided a opportunity for research projects, which would not be reasonable under „normal“ conditions.

The authors have done a great job in rainsing an interesting question at the right time. Their chosen methodology is sound, the analysis is profound and the conclusions are relevant. The discussion is balanced.

The only minimal issue is that the authors present the details of their respondents in the methods section. For me, they are part of the results (but this may reflect personal style).

In conclusion, I congratulate the authors to their brilliant work.

Reviewer #3: The authors investigated the relationship between social factors such as loneliness, social participation and satisfaction and pain intensity and disability in people with chronic pain who participated in an online survey during the time of COVID-19 related lockdowns. Although statistical analyses demonstrate associations between social factors and pain this study does not (due to its cross-sectional design) allow for any causal inferences. Nevertheless, the authors conclude that social factors can positively alter pain perception. Since these conclusions are not substantiated by the data, I cannot recommend the publication of this manuscript in its present form. However, the manuscript might be considered for publication after a major revision. Please see the concerns that need to be addressed below.

Major points

In some sections the language is unclear, making it difficult to follow. I advise the authors to thoroughly proof-read heir manuscript again or to obtain assistance of a copyeditor if needed.

Abstract:

The methods section in the abstract is confusing, please clarify that the interviews were conducted over a 38-day period, not the cross-sectional design.

The phrase “while pain management often focusses on the functional aspects of pain alleviation” does not belong into the results section but would rather be a point to mention in a discussion.

Conclusion:

I agree with the conclusion that increasing social satisfaction could be a point to consider in pain management, I do not see how this would be solved through technological applications. I thins this would rather call for social skills training or psychotherapy than for an app or technological gadget. I think the jump in logic from satisfaction with social roles to technological applications is a bit too far.

Methods:

Could you please elaborate on the procedure regarding the PROMIS Ability to Participate in Social Roles and Activities questionnaire? Which timeframe were participants asked to refer to when answering in the context of pre-COVID-19 and what was meant with prospectively (in the upcoming weeks, after the pandemic?). Given the well-known biases in retrospective reporting and in estimating future developments, why did you not assess the participation in social roles at the current time (i.e., at the point of the interview)?

Please elaborate on the criteria for chronic pain used in your study. How many participants were diagnosed with chronic pain by a medical professional and how many were self-diagnosed?

Data cleaning: Participants reporting that the questionnaires were boring does, by itself, not justify excluding them from data-analysis. Was there any objective reason to believe that their answers were not valid?

Were there any systematic differences between participants who completed all questionaries and those who quit early?

Table 1: Do these diagnoses include self-diagnoses? If so, it would be helpful to divide this table into self-diagnosed and diagnosed by medical professional.

Results:

Why are the data for the other sociodemographic questions such as fears surrounding COVID-19 and social distancing information not shown and analyzed?

Was the difference between pain intensity 7 days prior and at time of participation significant? Looking at the large standard deviations this seems unlikely that this difference of 1.5 points would be statistically significant. To me, this small difference seems overinterpreted.

Please refrain from interpretations of the data in the Results section.

Why did you not aggregate the values of “average, worst, and current” pain intensity to a composite score? This would yield a more robust outcome measure and fewer individual analyses.

Table 2: Please provide the possible range for each item. The term “Post-COVID-19” is confusing here because a) COVID-19 has not yet disappeared and b) in the methods it was described as “prospective”.

The language in the reporting of the results is very confusing. E.g., the authors talk about increases in loneliness predicting increases in pain interference. Since you had only one measurement timepoint, how could you measure “increases” in any of these items. Did you mean that greater loneliness predicted greater pain interference?

Discussion:

The authors make many claim that their data show that social satisfaction alters pain perception. However, the present data only demonstrate associations but no causations.

The claim “Our data however indicates that many of those with chronic pain were adjusting well to lockdowns and were coping well…” (ll.259-260) does not seem to be substantiated by the data. Which part of the results is this claim based on?

The claim “…our findings further emphasize that social factors can in fact positively alter pain experience” (ll.271-272) is not supported by your results, either. Correlation does not imply causation. Just the fact that there is an association between pain experience and social factors does not prove that social factors can alter the pain experience, it could also be the other way around.

Ll.279-280: These data do not show that satisfaction with participation in social roles reduces perceptions of chronic pain, they merely demonstrate an association.

The authors suggest that rather than trying to increase someone’s functional ability to actively socialize (e.g., going to a restaurant), a person with chronic pain could just use technology to obtain social satisfaction (ll.299-303). I think this conclusion is very premature since it does not consider the quality of the social experience and since it is not clear whether technologies provide the same level of satisfaction (consider e.g., the role of social touch in pain).

Unfortunately, the authors did not measure any (perceived) changes in social satisfaction, participation, or loneliness since or due to the pandemic. Therefore, it is not possible to draw any conclusions as to whether participants experienced any social changes for better or worse and whether potential social changes were related to potential changes in pain since/due to the pandemic. This limitation should be clearly pointed out in the Discussion.

Please comment on possible biases due to the sampling method used.

Minor points:

Please refer to “chronic pain patients” as “patients with chronic pain”. Since many of your participants were self-diagnosed it is not clear whether they were actual patients, referring to them as “people with (chronic) pain” would thus seem more appropriate.

For the PDI you mention a 5-point Likert scale ranging from 0 to 10. That would be an 11-point Likert scale.

Please adhere to common referencing style guidelines.

6. PLOS authors have the option to publish the peer review history of their article (what does this mean?). If published, this will include your full peer review and any attached files.

Reviewer #1: **Yes: **Carlo Matej RINAUDO

Reviewer #2: No

Reviewer #3: No

---

## [Author Response · Author response to Decision Letter 0]

1 Sep 2022

01 September 2022

Dear Mr Thomas Tischer,

Re: Social distancing with chronic pain during COVID-19: a cross-sectional correlational analysis.

Thank you for your helpful comments, as well as for the opportunity to revise this manuscript. As requested, this letter outlines how we have addressed each of the comments made. All changes made within the manuscript are given in red. 

We believe the paper is much stronger following these revisions . If we can be on any further help please do not hesitate to contact us.

 

Reviewer 1:

The authors thank reviewer 1 for taking the time to review this manuscript, they have addressed all comments as detailed in the responses below.

Grammatical Errors:

Line 99: "Thus, was reflective of the chronic pain population".  the subject is missing in this sentence. Maybe correct like this: "Thus, this sample was...".

Thank you for your comment, please find that Line 99 has now been changed to state:

“Thus, this sample was reflective of the chronic pain population …”

Line 230: "To examine for how social factors predicted pain during the first COVID-19 lockdown social variables...".  change sentence: "To examine how social factors predicted pain during the first COVID-19 lockdown, social variables...".

Thank you for your comment, please find that “for” has been removed and Line 230 now reads as:

“To examine how social factors predicted pain during the first COVID-19 lockdown, social variables …”

Line 263: "Importantly these confirm that those individuals who...".  the subject is missing in this sentence. Did the author mean: "Importantly this data confirms that the individuals who...".

Thank you for your comment, please find that Line 263 now reads as:

 “Importantly these data confirm that the individuals who …”

Line 312: there is a question mark that shouldn't be there.

Thank you for your comment, the additional question mark has now been removed from Line 312.

Abstract:

1) I would suggest taking the last sentence of the results (line 41-43) and including it in the conclusions. After that I would suggest developing a bit more the results.

Thank you for your comment, Line 41-43 has been added to the conclusions, therefore the abstract conclusion now reads as:

“Therefore, while pain management often focuses on the functional aspects of pain alleviation, these findings suggest psychological aspects of socialising satisfaction also impact pain experience. Pain management strategies should consider ways to increase social satisfaction in individuals with chronic pain, perhaps by facilitating socialisation in the home environment through technological applications.”

Further, abstract results have been developed further and now reads as:

“Multiple regression analysis revealed social satisfaction significantly predicted pain experience, with a reduction in social participation during COVID-19 lockdowns increasing pain disability, and increased social satisfaction associated with decreasing pain intensity”

Methods:

2) Line 93-94: "...having experienced consistent pain for the past 6 months..."  maybe specify if was there a cut-off for pain intensity in this criteria.

Thank you for your comment, whilst there was no explicit cut-off, it was intended to be 6 months prior to agreeing to participate. This has been added for clarity to Line 93-94 as shown below:

“… having experienced consistent pain for the past 6 months at time of agreeing to participate …”

Results:

3) Line 222-224: "In our study we observe a marginal decrease in average pain intensity 7-days prior to (...) and pain intensity at the time of participation..."  marginal decrease compared to what? Is it between average pain intensity and pain intensity at the time of participation? In that case I suggest modifying it as follows: "In our study we observe a marginal decrease between average pain intensity 7-days prior to (...) and pain intensity at the time of participation...". Also, although it's descriptive statistics, it should be specified whether this difference is statistically significant or not.

Thank you for your comment, we have added statistical significance and altered the sentence structure accordingly:

“Despite observing a marginal decrease in average pain intensity 7-days prior to participation (M = 57.68, SD = 19.73) and pain intensity at time of participation (M = 56.02, SD = 23.07), this change was not statistically significant (p = .33).”

Conclusions:

4) Beware that the chronic pain population of this study is fairly heterogeneous (as shown by the diagnostic table S1). This should be noted in the conclusions and taken into consideration when stating recommendations/suggestions or making conclusions (such as Line 277-278 and Line 292-295).

Thank you for your comment, we have added the following point at the end of Line 292-295’s paragraph:

“However, given the heterogeneity of the current sample, efficacy of the latter may differ dependent on, for example, pain condition, age, gender, etc.”

Reviewer 2: 

Comments to the Author:

The authors present an extremely well written manuscript an the effect of social distancing on people with chronic pain during the COVID lockdown. The COVID pandemic and the social restrictions provided an situation which was unprecedented before. Of course, this provided a opportunity for research projects, which would not be reasonable under „normal“ conditions.

The authors have done a great job in raising an interesting question at the right time. Their chosen methodology is sound, the analysis is profound and the conclusions are relevant. The discussion is balanced … In conclusion, I congratulate the authors to their brilliant work.

Thank you very much for your positive feedback, the authors are grateful that you found the article enjoyable and value the findings so highly.

Methods/Results:

The only minimal issue is that the authors present the details of their respondents in the methods section. For me, they are part of the results (but this may reflect personal style).

Thank you for your comment, please find Line 99-113 of the original methods section have been moved to the beginning of the results section.

Reviewer 3: 

Comments to the Author:

The authors investigated the relationship between social factors such as loneliness, social participation and satisfaction and pain intensity and disability in people with chronic pain who participated in an online survey during the time of COVID-19 related lockdowns. Although statistical analyses demonstrate associations between social factors and pain this study does not (due to its cross-sectional design) allow for any causal inferences. Nevertheless, the authors conclude that social factors can positively alter pain perception. Since these conclusions are not substantiated by the data, I cannot recommend the publication of this manuscript in its present form. However, the manuscript might be considered for publication after a major revision. Please see the concerns that need to be addressed below.

Thank you for taking the time to review this manuscript, the authors value your feedback and have made updates in line with your comments.

Major points:

In some sections the language is unclear, making it difficult to follow. I advise the authors to thoroughly proof-read their manuscript again or to obtain assistance of a copyeditor if needed.

Thank you for raising this point, the authors have thoroughly proof-read the article and rectified any language or grammatical errors which previously hindered readability.

Abstract:

1) The methods section in the abstract is confusing, please clarify that the interviews were conducted over a 38-day period, not the cross-sectional design.

Thank you for your comment, the methods section of the abstract has now been amended to reflect your point:

“In a cross-sectional correlational design, questionnaire data was collected over a 38-day period during the March 2020 COVID-19 lockdown …”

2) The phrase “while pain management often focusses on the functional aspects of pain alleviation” does not belong into the results section but would rather be a point to mention in a discussion.

Thank you for your comment, Line 41-43 has been added to the conclusions, therefore the abstract conclusion now reads as:

“Therefore, while pain management often focuses on the functional aspects of pain alleviation, these findings suggest psychological aspects of socialising satisfaction also impact pain experience. Pain management strategies should consider ways to increase social satisfaction in individuals with chronic pain, perhaps by facilitating socialisation in the home environment through technological applications.”

3) I agree with the conclusion that increasing social satisfaction could be a point to consider in pain management, I do not see how this would be solved through technological applications. I think this would rather call for social skills training or psychotherapy than for an app or technological gadget. I think the jump in logic from satisfaction with social roles to technological applications is a bit too far.

Thank you for your comment, the idea of utilising technological applications is to adhere to the fact such applications were utilised frequently during the COVID-19 lockdown as a means to socialise. The use of social skills training or psychotherapy would suggest individuals with chronic pain have some sort of socialising deficit to improve on; that is not what the authors are suggesting. Instead the aim is to facilitate the ability to socialise with the external world, whilst in the home environment. The use of these outside of the lockdown would allow individuals to socialise in the same manner, and continue to improve their social satisfaction. 

The authors have removed the term “using these technological applications” from the abstract conclusion so that it now reads as:

“Pain management strategies should consider ways to increase social satisfaction in individuals with chronic pain, perhaps by facilitating socialisation in the home using remote communication methods similar to those utilised during the COVID-19 lockdown”

Methods:

1) Could you please elaborate on the procedure regarding the PROMIS Ability to Participate in Social Roles and Activities questionnaire? Which timeframe were participants asked to refer to when answering in the context of pre-COVID-19 and what was meant with prospectively (in the upcoming weeks, after the pandemic?). Given the well-known biases in retrospective reporting and in estimating future developments, why did you not assess the participation in social roles at the current time (i.e., at the point of the interview)?

Thank you for your comment, we have rephrased this section to reflect your queries. Pre-COVID 19 could have been any time prior to COVID-19 lockdown initiation, and prospectively has now been reflective with the term post-COVID-19 lockdown initiation (which would be current time/point of interview).

“The question bank was answered twice in two different contexts: first in the context of pre-COVID-19 lockdown initiation, second post-COVID-19 lockdown initiation.”

2) Please elaborate on the criteria for chronic pain used in your study. How many participants were diagnosed with chronic pain by a medical professional and how many were self-diagnosed?

Thank you for your query, criteria for chronic pain was either a diagnosis of chronic pain or pain that had lasted a minimum of 6 months . Due to the self-selecting nature of the study, we did not have ethical approval to confirm via medical notes or a medical health care professional a formal chronic pain diagnosis. However, the use of identifying pain for the past 6 months was intended to capture a population whom, if not diagnosed by a medical health care professional, satisfied the basic diagnostic criteria. Further, our question asked participants to only identify their diagnoses from the list. Those who did not have a formal chronic pain diagnosis (n = 2) were asked to select a different option to indicate this.

3) Data cleaning: Participants reporting that the questionnaires were boring does, by itself, not justify excluding them from data-analysis. Was there any objective reason to believe that their answers were not valid? Were there any systematic differences between participants who completed all questionaries and those who quit early?

Thank you for your comment, participants who indicated the study was boring further indicated to ignore their answers indicating they had no interest in the study so were mindlessly clicking responses. Therefore, as indicated, their responses were not influenced by their own experiences their data was excluded to ensure validity of responses.

4) Table 1: Do these diagnoses include self-diagnoses? If so, it would be helpful to divide this table into self-diagnosed and diagnosed by medical professional.

These diagnoses do not include participants who classified they did not have a diagnosis but had recurring pain for the past 6 months (n = 2) as it was assumed they were self-diagnosed given they stated they had no diagnosis.

Results:

1) Why are the data for the other sociodemographic questions such as fears surrounding COVID-19 and social distancing information not shown and analyzed?

Thank you for your comment, this data was not deemed to add additional relevance to the manuscript given many did not provide any insight into the social factors of pain which were core to the current manuscript. 

2) Was the difference between pain intensity 7 days prior and at time of participation significant? Looking at the large standard deviations this seems unlikely that this difference of 1.5 points would be statistically significant. To me, this small difference seems overinterpreted.

Thank you for your comment, we have added statistical significance and altered the sentence structure accordingly:

“Despite observing a marginal decrease in average pain intensity 7-days prior to participation (M = 57.68, SD = 19.73) and pain intensity at time of participation (M = 56.02, SD = 23.07), this change was not statistically significant (p = .33).”

3) Please refrain from interpretations of the data in the Results section.

Thank you for highlighting this, we have removed any notable interpretations from the results section.

E.g. “… suggest changes to social dynamics during COVID-19 may positively impact intrinsic chronic pain perception during the early stages of a lockdown. Thus, further indicating social interaction may be an important factor in reducing the impact of pain.”

4) Why did you not aggregate the values of “average, worst, and current” pain intensity to a composite score? This would yield a more robust outcome measure and fewer individual analyses.

Thank you for your query, given the study was structured around specific timepoints of COVID-19 it was decided not to aggregate scores to identify how effects of discrete timepoints that were reflective of the changing timepoints of the COVID-19 lockdown. For example, here we have measured 3 differing experiences: How intense has pain been at its worst shows, even if brief maybe the most extreme response to the pandemic, current pain shows influence on responses at the time and average pain already shows a general response to pain. By aggregating a composite score the authors believe these experiences will have been generalised as opposed to giving a clear and distinct picture of 3 separate pain experiences,

5) Table 2: Please provide the possible range for each item. The term “Post-COVID-19” is confusing here because a) COVID-19 has not yet disappeared and b) in the methods it was described as “prospective”.

Thank you for your comment, context for the term has been added in both the methods and Table 2:

Methods: “The question bank was answered twice in two different contexts: first in the context of pre-COVID-19 lockdown initiation, second post-COVID-19 lockdown initiation.”

Table 2: “Difference in Ability to Participate in Social Roles and Activities Pre- and Post-COVID-19 lockdown initiation”

6) The language in the reporting of the results is very confusing. E.g., the authors talk about increases in loneliness predicting increases in pain interference. Since you had only one measurement timepoint, how could you measure “increases” in any of these items. Did you mean that greater loneliness predicted greater pain interference?

Thank you for highlighting this, please fine Line 257-259 now reads as follows:

“… Examination of the individual predictors revealed reports of higher loneliness significantly predicted greater pain interference (t = 3.10, p = .003) while increases in social role satisfaction significantly predicted lower pain interference (t = 6.16, p < .001) …”

Discussion:

1) The authors make many claim that their data show that social satisfaction alters pain perception. However, the present data only demonstrate associations but no causations.

Thank you for your comment, the authors have altered wording throughout the discussion to reflect association rather than causation.

2) The claim “Our data however indicates that many of those with chronic pain were adjusting well to lockdowns and were coping well…” (ll.259-260) does not seem to be substantiated by the data. Which part of the results is this claim based on?

Thank you for pointing this out, we have added the following point based upon the social satisfaction data:

“Our data indicates that many living with chronic pain were more satisfied with their ability to socialize during, compared to prior to, the lockdown. “ 

3) The claim “…our findings further emphasize that social factors can in fact positively alter pain experience” (ll.271-272) is not supported by your results, either. Correlation does not imply causation. Just the fact that there is an association between pain experience and social factors does not prove that social factors can alter the pain experience, it could also be the other way around.

Thank you for highlighting this, this sentence has been rephrased to reflect this association rather than causation:

“Yet, despite suggested negative social experiences accompanying chronic pain, our findings further emphasise that social factors may in fact positively alter pain experience (7, 14, 21-23)”

4) Ll.279-280: These data do not show that satisfaction with participation in social roles reduces perceptions of chronic pain, they merely demonstrate an association.

Thank you for highlighting this, this sentence has been rephrased to reflect this associated rather than causation:

“Our most consistent finding here was that satisfaction with participation in social roles and activities may reduce perceptions of chronic pain.”

5) The authors suggest that rather than trying to increase someone’s functional ability to actively socialize (e.g., going to a restaurant), a person with chronic pain could just use technology to obtain social satisfaction (ll.299-303). I think this conclusion is very premature since it does not consider the quality of the social experience and since it is not clear whether technologies provide the same level of satisfaction (consider e.g., the role of social touch in pain).

Thank you for your comment, the idea of utilising technological applications is to adhere to the fact such applications were utilised frequently during the COVID-19 lockdown as a means to socialise. The use of these outside of the lockdown would allow individuals to socialise in the same manner, and continue to improve their social satisfaction. However, the authors appreciate this comment and have suggested further methods of increasing social satisfaction:

“Therefore, it is plausible if an individual is unable to leave their house at any given point due to their pain intensity, rather than increasing functional abilities to actively socialise by physically going to a restaurant with someone helping the individual travel safely, for example, ways to bring social satisfaction into the immediate home, through technology, may be an and effective alternative to change chronic pain experiences.”

6) Unfortunately, the authors did not measure any (perceived) changes in social satisfaction, participation, or loneliness since or due to the pandemic. Therefore, it is not possible to draw any conclusions as to whether participants experienced any social changes for better or worse and whether potential social changes were related to potential changes in pain since/due to the pandemic. This limitation should be clearly pointed out in the Discussion.

Thank you for your comment, please find the following point that has been added to the discussion:

“Moreover, variables such as loneliness and social satisfaction were only measured at time of participation. Therefore, improvements or detriments to loneliness and social satisfaction as a consequence of the to the pandemic lockdown cannot be determined.”

7) Please comment on possible biases due to the sampling method used.

Thank you for your comment, the authors have added the following statement to the discussion:

“Additionally, use of snowball sampling may have contributed to potential recruitment bias given a network of individuals who lead back to the principal investigator are targeted, creating a sample which by definition cannot be entirely random and not representative of the chronic pain population (35).”

Minor points:

1) Please refer to “chronic pain patients” as “patients with chronic pain”. Since many of your participants were self-diagnosed it is not clear whether they were actual patients, referring to them as “people with (chronic) pain” would thus seem more appropriate.

Thank you for your comment, this has been amended as “individuals with chronic pain” throughout the manuscript.

2) For the PDI you mention a 5-point Likert scale ranging from 0 to 10. That would be an 11-point Likert scale.

Thank you for your comment, please find Line 136-137 now reads as:

“… The PDI asks participants to rate on a 11-point Likert scale ranging from 0 (no disability) to 10 (worst disability) …”

3) Please adhere to common referencing style guidelines.

Thank you for your comment, referencing style guidelines have been reviewed and altered where fit.

---

## [Editor Report · Decision Letter 1]

21 Sep 2022

Social distancing with chronic pain during COVID-19: a cross-sectional correlational analysis.

PONE-D-22-02463R1

Dear Dr. Donaghy,

We’re pleased to inform you that your manuscript has been judged scientifically suitable for publication and will be formally accepted for publication once it meets all outstanding technical requirements.

Kind regards,

Helmar Bornemann-Cimenti, M.D., D.Med.Sci., M.Sc., MBA

Guest Editor

PLOS ONE

Additional Editor Comments (optional):

The authors present a revised version of their originial submission, whicht fully addresses all issues raised by the reviewers. The changes further improved the clarity af the manuscript, which is now acceptable for publication in PLOS ONE.

Disclosure: I participated as a reviewer for the initial evaluation of this manuscript.